# Hybrid Evolutionary Algorithm with Optimized Operators for Scheduling on Permutations

## Abstract

New evolutionary algorithm with optimized operators is proposed for a scheduling problem on permutations. The optimal recombination problem is solved in a crossover operator. The computational complexity of this problem is analyzed. Adaptive mutation and local search heuristics are integrated in our algorithm. We construct the initial population by means of greedy constructive heuristics. A computational experiment on the OR-Library instances shows that the proposed algorithm yields results competitive to those of well-known algorithms and confirms that the optimal recombination may be used successfully in evolutionary algorithms.

## 1 Introduction

### 1.1 Problem Statement

We consider the following problem of scheduling a set of jobs $\mathcal{J} = \{1, \ldots, n\}$ on a single machine. Job $j \in \mathcal{J}$ is characterized by release date $r_j$, processing time $p_j$, positive weight $w_j$, and due date $d_j$ by which it should be finished. The machine can execute at most one job at a time and preemptions are disallowed. Let $C_j$ denote the completion time of job $j \in \mathcal{J}$, then the tardiness $T_j$ of job $j$ is computed as $\max\{0; C_j - d_j\}$. The goal is to construct a schedule such that the total weighted tardiness $\sum_{j \in \mathcal{J}} w_j T_j$ is minimized. Using the classical three field notation our problem is denoted as $1|r_j, d_j, w_j| \sum_j w_j T_j$ and called *the one-machine total weighted tardiness problem (TWTP)*. This problem arises in several practical settings, in particular in the chemical industry Wagner et al. (2002). In addition, scheduling models with a single machine also have implications for scheduling research involving multiple machines, where the results obtained from single-machine problems can often be applied to more complex scheduling environments, such as parallel machines, flow shops, job shops, and open shops.

Let $\pi = (\pi_1, \ldots, \pi_n)$ denote a permutation of the jobs, i.e. $\pi_i$ is the $i$-th job on the machine, $i = 1, \ldots, n$. Then the completion times $C(\pi_i) := C(\pi_{i-1}) + p_{\pi_i}$, and tardiness $T(\pi_i) := \max\{0; C(\pi_i) - d_{\pi_i}\}$ for jobs in positions $i = 1, \ldots, n$ (suppose $C(\pi_0) := 0$). We denote the total weighted tardiness for permutation $\pi$ by $T(\pi) = \sum_{i=1}^{n} w_{\pi_i} T(\pi_i)$.

The considered problem is NP-hard, so exact techniques are only applicable to small-sized problems in practice. Therefore metaheuristics are appropriate for the problem, in particular evolutionary algorithms (EAs).

Performance of evolutionary algorithms depends on the crossover operator, where the components of parent solutions are combined to build the offspring. *Optimal recombination problem* (ORP) is a subproblem, that consists in finding the best possible offspring as a result of a crossover operator, given two feasible parent solutions under the requirement that the recombination should be respectful and gene transmitting as coined by N. Radcliffe Radcliffe (1994).

The optimal recombination problem for $1|r_j, d_j, w_j| \sum_j w_j T_j$ with position-based representation of solutions is formulated as follows. Given two parent solutions $\pi^1$ and $\pi^2$. It is required to find a permutation $\pi'$ such that:

(I) $\pi'_i = \pi^1_i$ or $\pi'_i = \pi^2_i$ for all $i = 1, \ldots, n$;

(II) $\pi'$ has the minimum value of objective function $T(\pi')$ among all permutations that satisfy condition (I).

The corresponding recombination operators, where the ORP is solved, are called optimized crossovers. Previously, the computational complexity of optimal recombination of single-machine scheduling problem with setup times Eremeev & Kovalenko (2012), permutation flow-shop scheduling problems Kovalenko (2016), Travelling Salesmen Problem (TSP) and Boolean programming problems Eremeev & Kovalenko (2014) has been investigated. It is known algorithms of solving NP-hard ORPs for permutation problems and their variations (e.g., dynamic programming Yagiura & Ibaraki (1996), enumeration of prefect matchings in a special bipartite graph Kovalenko (2016), partition of graph-vertices into recombining components Tinos et al. (2020), analog of the recursive algorithm of Eppstein for cubic graphs Eremeev & Kovalenko (2020)). Experimental evaluation shows that the optimal recombination can be successfully used in EAs (see, e.g., Eremeev & Kovalenko (2016); Yagiura & Ibaraki (1996); Balas & Niehaus (1998)).

In this paper we propose new evolutionary algorithm with optimal recombination for the single-machine total weighted tardiness problem. The optimal recombination problem is solved in a crossover operator. The computational complexity of this problem is analysed. We construct the initial population means of greedy constructive heuristics. The insert and swap local search heuristics are used to improve the initial and the final populations. A computational experiment on the OR-Library instances shows that the proposed algorithm yields competitive results.

## 1.2 Previous Research

Now we briefly present the state-of-the-art metaheuristics known for problem $1|r_j, d_j, w_j| \sum_j w_j T_j$: local search methods, ant colony optimization methods, evolutionary algorithms and other population-based methods.

Good experimental results for $1|r_j, d_j, w_j| \sum_j w_j T_j$ are demonstrated by single-solution and population-based local search methods Wang & Tang (2009); Geiger M.J. (2010); Yahyaoui et al. (2013); Fu & Chung (2016); Congram et al. (2002); Chunga et al. (2017); Bilge et al. (2007). The reason is the "good" properties of the objective function, which significantly reduce the search area of solutions in the neighborhoods (see, e.g., Geiger M.J. (2010)).

Ant colony optimization (ACO) is inspired by the behavior of real ant colonies. The main steps of the ACO are constructing solutions, local improving and updating trails. In den Besten et al. (2000); Anghinolfi & Paolucci (2008); Abitz et al. (2020), competitive ACO algorithms have been developed for the TWTP.

Hybrid evolutionary algorithms (HEAs) occupy one of the leading positions among the metaheuristics known for the considered problem. One of the first genetic algorithms for the TWTP was presented in Avci et al. (2003). The algorithm uses various greedy heuristics in operators assigning jobs to positions. The authors applied uniform and single-point crossovers, and perturbation techniques. Different crossover operators were compared in Kellegöz et al. (2008). The best results in the canonical genetic algorithm (basic scheme) for the TWTP were demonstrated by position and order based crossovers. L. Chaabane Bouchra & Chaabane (2007) has proposed for TWTP a hybrid genetic algorithm applying simulated annealing at local improving stage, one point crossover and insert mutation. The hybrid method Ding et al. (2017) combining evolutionary algorithm with position-based crossover and dynasearch procedure demonstrates the most effective results at this time in terms of both objective value and running time.

In Differential evolution (DE) methods Nearchou (2012); Tasgetiren et al. (2004; 2006) for the considered problem, a solution is encoded by a real vector of length $n$, and the corresponding permutation is constructed by means of the smallest position value (SPV) rule, where jobs are sorted in increasing order of vector-component values. Various local search methods (for example, insert and swap) and their combinations are used at iterations to improve results.

Bee colony algorithm is also a population-based method inspired by the nectar searching behaviors of bees and their activities in the hive. In the BCO algorithm from Sharma et al. (2019) for the considered problem the solutions are represented by real-valued vectors, and such vectors are decoded

into permutations using the smallest position value (SPV) rule. The operators from continuous optimization and local search improving methods are involved in the search mechanism.

One of the aim of this paper is to compare the optimized crossover based on position solution representation with other known techniques, in particular local optimization mathods. As far as we know, such results have not provided in the literature at this time.

## 2 COMPUTATIONAL COMPLEXITY OF OPTIMAL RECOMBINATION PROBLEM

For proving NP-hardness of the ORP for $1|r_j, d_j, w_j| \sum_j w_j T_j$ we will use the following *Restricted Even-Odd Partition Problem* Du & Leung (1990): given ordered set $A = \{a_1, a_2, \ldots, a_{2n_0}\}$ and weight $e_i$ of each element $a_i \in A$, where

$$\sum_{a_i \in A} e_i = 2E, \ e_i > e_{i+1}, \ i = 1, \ldots, 2n_0 - 1, \tag{1}$$

$$e_{2j} > e_{2j+1} + \delta, \ e_{2j-1} \leq e_{2j} + \delta, j = 1, \ldots, n_0. \tag{2}$$

Here $\delta = \frac{1}{2} \sum_{i=1}^{n_0} (e_{2i-1} - e_{2i})$. Note that $E = \frac{1}{2} \sum_{i=1}^{2n_0} e_i = \sum_{i=1}^{n_0} e_{2i-1} - \delta = \sum_{i=1}^{n_0} e_{2i} + \delta$. The question is to decide whether set $A$ can be partitioned on two subsets $A_1$ and $A_2$ such, that

$$\sum_{a_i \in A_1} e_i = \sum_{a_i \in A_2} e_i = E, \ |A_1| = |A_2| = n_0$$

and subset $A_1$ includes only one element from each pair $a_{2i-1}, \ a_{2i}$.

We prove NP-hardness of the ORP. An exact method for the ORP and polynomial solvability in "almost all" cases will be presented and analyzed.

### 2.1 NP-HARDNESS

**Theorem 1** *Optimal recombination problem (I)-(II) for $1|r_j = 0, d_j| \sum_j T_j$ is $NP$-hard.*

**Proof.** The presented reduction uses some ideas from Du & Leung (1990), but we prove NP-hardness of new ORP of scheduling problem and identify its property instead of the classic scheduling statement. Let the number of jobs $n = 3n_0 + 1$. The jobs $j \in \{1, \ldots, 2n_0\}$ will be called $S$-jobs: $S_1, \ldots, S_{2n_0}$, and the rest $n_0 + 1$ will be called $B$-jobs: $B_1, \ldots, B_{n_0}, B_{n_0+1}$. We denote by $b$ the value $(4n_0 + 1)\delta$.

Processing times of $S$-jobs are equal to $p(S_i) = e_i, \ i = 1, \ldots, 2n_0$, while durations of $B$-jobs $p(B_i) = b, i = 1, \ldots, n_0 + 1$. Now we provide due dates of jobs

$$d(S_i) = \begin{cases} (j-1)b + \delta + (e_2 + e_4 + \cdots + e_{2j}), & \text{if } i = 2j - 1, \\ d(S_{2j-1}) + 2(n_0 - j + 1)(e_{2j-1} - e_{2j}), & \text{if } i = 2j, \end{cases}$$

$j = 1, \ldots, n_0$.

$$d(B_i) = \begin{cases} ib + (e_2 + e_4 + \cdots + e_{2i}), & \text{if } i = 1, \ldots, n_0, \\ d(B_{n_0}) + \delta + b, & \text{if } i = n_0 + 1. \end{cases}$$

Define the parent permutations in ORP as

$$\pi^1 := (S_1, B_1, S_3, B_2, \ldots, B_{n_0-1}, S_{2n_0-1}, B_{n_0}, B_{n_0+1}, S_{2n_0}, S_{2n_0-2}, \ldots, S_2),$$

$$\pi^2 := (S_2, B_1, S_4, B_2, \ldots, B_{n_0-1}, S_{2n_0}, B_{n_0}, B_{n_0+1}, S_{2n_0-1}, S_{2n_0-3}, \ldots, S_1).$$

Show that there is a permutation $\pi'$ satisfying conditions (I) and $\sum_{j=1}^{n} T(\pi'_j) \leq T_0$, if and only if the Restricted Even-Odd Partition Problem has a positive answer. Here $T_0 := E + n_0 P - n_0(n_0 - 1)b/2 - n_0\delta - \sum_{i=1}^{n_0}(n_0 - i + 1)(e_{2i-1} + e_{2i})$, where $P := (n_0 + 1)b + 2E$ is the total duration of all jobs.

We start the proof with the statement that the inequality $\sum_{j=1}^{n} T(\pi'_j) \geq T_0$ holds for permutation

$$\pi' := (S_{1,1}, B_1, S_{2,1}, B_2, \ldots, B_{n_0-1}, S_{n_0,1}, B_{n_0}, B_{n_0+1}, S_{n_0,2}, S_{n_0-1,2}, \ldots, S_{1,2})$$

(here $\{S_{i,1},\ S_{i,2}\} = \{S_{2i-1},\ S_{2i}\}$). Moreover, $\sum_{j=1}^{n} T(\pi'_j) = T_0$ if and only if $\sum_{i=1}^{n_0} p(S_{i,1}) = \sum_{i=1}^{n_0} p(S_{i,2})$.

Compute the total tardiness for jobs $B_i$, $i = 1, \ldots, n_0$. Since $e_{2j-1} > e_{2j}$, then $C(B_i) \geq d(B_i)$ for completion times of jobs $B_i$, $i = 1, \ldots, n_0$. In addition $C(B_i) = ib + \sum_{j=1}^{i} p(S_{j,1})$, $\sum_{i=1}^{n_0} d(B_i) = \frac{n_0(n_0+1)}{2} b + \sum_{i=1}^{n_0} (n_0 - i + 1)e_{2i}$. Therefore

$$\sum_{i=1}^{n_0} (C(B_i) - d(B_i)) = \sum_{i=1}^{n_0} C(B_i) - \sum_{i=1}^{n_0} d(B_i) =$$

$$\frac{n_0(n_0+1)}{2} b + \sum_{i=1}^{n_0} (n_0 - i + 1)p(S_{i,1}) - \frac{n_0(n_0+1)}{2} b - \sum_{i=1}^{n_0} (n_0 - i + 1)e_{2i} =$$

$$\sum_{i=1}^{n_0} (n_0 - i + 1)\left(p(S_{i,1}) - e_{2i}\right).$$

Now we calculate the total tardiness for jobs $S_{i,2}$, $i = 1, \ldots, n_0$. Note that we have $C(S_{i,2}) \geq d(S_{i,2})$ for jobs $S_{i,2}$, $i = 1, \ldots, n_0$, because $C(S_{i,2}) \geq (n_0 + 1)b + \sum_{j=1}^{n_0} a_{2j}$, but $d(S_{i,2}) \leq (n_0 - 1)b + (2n_0 + 1)\delta + \sum_{j=1}^{n_0} a_{2j}$ from definitions of due dates and processing times. Moreover $C(S_{i,2}) = P - \sum_{j=1}^{i-1} p(S_{j,2})$. If $S_{i,2} = S_{2i-1}$, then $d(S_{i,2}) = (i-1)b + \delta + \sum_{j=1}^{i} e_{2j}$, while if $S_{i,2} = S_{2i}$, then $d(S_{i,2}) = (i - 1)b + \delta + \sum_{j=1}^{i} e_{2j} + 2(n_0 - i + 1)(e_{2i-1} - e_{2i})$. Therefore $d(S_{i,2}) = (i-1)b + \delta + \sum_{j=1}^{i} e_{2j} + (n_0 - i + 1)(e_{2i-1} - e_{2i}) + (n_0 - i + 1)\left(p(S_{i,1}) - p(S_{i,2})\right)$. Thus,

$$\sum_{i=1}^{n_0} (C(S_{i,2}) - d(S_{i,2})) =$$

$$n_0 P - \sum_{i=1}^{n_0-1} (n_0 - i)p(S_{i,2}) - \frac{n_0(n_0 - 1)}{2} b - n_0\delta -$$

$$\sum_{i=1}^{n_0} (n_0 - i + 1)e_{2i-1} - \sum_{i=1}^{n_0} (n_0 - i + 1)\left(p(S_{i,1}) - p(S_{i,2})\right) =$$

$$n_0 P - \frac{n_0(n_0 - 1)}{2} b - n_0\delta - \sum_{i=1}^{n_0} (n_0 - i + 1)e_{2i-1} - \sum_{i=1}^{n_0} (n_0 - i + 1)p(S_{i,1}) + \sum_{i=1}^{n_0} p(S_{i,2}).$$

We have the total tardiness $\sum_{j=1}^{n} T(\pi'_j)$ over all jobs equal to

$$n_0 P - \frac{n_0(n_0 - 1)}{2} b - n_0\delta - \sum_{i=1}^{n_0} (n_0 - i + 1)\left(e_{2i-1} + e_{2i}\right) +$$

$$\sum_{i=1}^{n_0} p(S_{i,2}) + \sum_{i=1}^{n_0} T(S_{i,1}) + T(B_{n_0+1}).$$

If $\sum_{i=1}^{n_0} p(S_{i,1}) \leq E = \sum_{i=1}^{n_0} e_{2i} + \delta$, then for any $k = 1, \ldots, n_0$ we have $\sum_{i=1}^{k} p(S_{i,1}) \leq \sum_{i=1}^{k} e_{2i} + \delta$ (proved by contradiction). And form the definition of values $d(S_i)$ and $d(B_{n_0+1})$ we obtain $C(S_{i,1}) \leq d(S_{i,1})$ and $T(S_{i,1}) = 0$; $C(B_{n_0+1}) \leq d(B_{n_0+1})$ and $T(B_{n_0+1}) = 0$. From the definition of $T_0$ and expression for $\sum_{j=1}^{n} T(\pi'_j)$ we conclude that $\sum_{i=1}^{n_0} p(S_{i,2}) \geq E$ and $\sum_{j=1}^{n} T(\pi'_j) \geq T_0$. Moreover, the equality is possible only when $\sum_{i=1}^{n_0} p(S_{i,1}) = \sum_{i=1}^{n_0} p(S_{i,2}) = E$.

If $\sum_{i=1}^{n_0} p(S_{i,1}) > E$, then

$$T(B_{n_0+1}) = \sum_{i=1}^{n_0} p(S_{i,1}) + (n_0 + 1)b - \left((n_0 + 1)b + \delta + \sum_{i=1}^{n_0} e_{2i}\right) = \sum_{i=1}^{n_0} p(S_{i,1}) - E.$$

Moreover, we can find index $l$ such that $\sum_{i=1}^{l} p(S_{i,1}) > \sum_{i=1}^{l} e_{2i} + \delta$. Let $k$ is the smallest from such indices, then $S_{k,1} = S_{2k-1}$, $C(S_{k,1}) > d(S_{k,1}) = \sum_{i=1}^{k} e_{2i} + \delta + (k-1)b$ by definition and $T(S_{k,1}) > 0$. As a result we have $\sum_{j=1}^{n} T(\pi'_j) \geq$

$$n_0 P - \frac{n_0(n_0 - 1)}{2}b - n_0\delta - \sum_{i=1}^{n_0}(n_0 - i + 1)(e_{2i-1} + e_{2i}) +$$

$$\left(\sum_{i=1}^{n_0} p(S_{i,2}) + \sum_{i=1}^{n_0} p(S_{i,1}) - E\right) + T(S_{k,1}) > T_0.$$

So, a permutation $\pi'$ satisfying conditions (I) and $\sum_{j=1}^{n} T(\pi'_j) \leq T_0$ exists if and only if the Restricted Even-Odd Partition Problem has a positive answer. The reduction is polynomial.

## 2.2 EXACT METHOD

For solving the ORP of $1|r_j, d_j, w_j| \sum_j w_j T_j$ we will use the following exact method.

Construct a bipartite graph $G = (\mathcal{J}_n, \mathcal{J}, U)$ with subsets of vertices $\mathcal{J}_n, \mathcal{J}$ having equal sizes and the set of edges $U = \{(i, j) : i \in \mathcal{J}_n, j = \pi_i^1 \text{ or } j = \pi_i^1\}$, where $\mathcal{J}_n = \{1, \ldots, n\}$. There exists a one-to-one correspondence between the set of feasible solutions to the considered ORP and the set of perfect matchings in graph $\bar{G}$: permutation $\pi^m = (j^1, \ldots, j^n)$ gives perfect matching $m^\pi = \{\{1, j^1\}, \ldots, \{n, j^n\}\}$ and vice versa.

An edge $\{i, j\} \in \bar{U}$ is called *special* if $\{i, j\}$ belongs to any perfect matching of $G$. The maximum connected subgraph of $G$ containing at least two edges represents a cycle. Denote by $q$ the number of cycles in $G$. Each cycle $G$ contains exactly two maximal (perfect) matchings, and they are edge disjoint. Moreover, any perfect matchings in $\bar{G}$ is uniquely defined by maximal matchings of cycles and special edges. Note that maximal matchings of cycles and special edges may be computed in $O(n)$ time.

Thus, ORP (I)-(II) can be solved by enumeration of perfect matchings in graph $G$. For each perfect matching $m$ we construct the corresponding permutation $\pi$ and calculate the objective value. As a result, we find the optimal solution of the ORP in $O(2^q n)$ time, where $q \leq \frac{n}{2}$. Moreover, "almost all" pairs of parent permutations give graphs $G$ with $q \leq \frac{\ln(n)}{\ln(2)}$ cycles, i.e., ORP (I)-(II) has at most $n$ feasible solutions (see proof in Eremeev & Kovalenko (2012)).

In addition to the previous research we propose here the following speed-up procedure for the considered ORP based on the properties of criterion $\sum_{j \in \mathcal{J}} w_j T_j$. In the searching process we will guarantee that the maximum matching is changed only in one cycle (for example, using the code of Gray), when we go from one perfect matching to another. Let permutation $\pi$ (corresponding to the perfect matching $m$ at the previous step) is replaced by permutation $\pi' = (\pi_1, \ldots, \pi'_a, \pi'_{a+1}, \ldots, \pi'_b, \ldots, \pi_n)$ by changing the maximum matching in the cycle with the minimum index of the left part equal to $a$ and the maximum index equal to $b$. Then, the completion times of jobs in positions $a, a + 1, \ldots, b$ are only changed for the permutation $\pi'$ in comparison to the permutation $\pi$. Only these modifications should be taken into account in computing the objective value for $\pi'$ using the objective value for $\pi$ (if $r_j$ is nonzero, then the completion times of jobs in positions $b, b + 1, \ldots, n$ must be updated after changing even one maximum matching of a cycle). Experimental evaluation of the proposed speed-up procedure is given in Section 4.

Solving Optimal recombination problem in the crossover operator may be considered as a derandomization of the well-known *CX* (Cycle Crossover) operator Kellegöz et al. (2008), where maximal matchings are selected randomly in cycles.

## 3 HYBRID EVOLUTIONARY ALGORITHM

We propose hybrid evolutionary algorithm with optimal recombination based on the steady state replacement scheme. Each individual corresponds to the tentative solution of the problem (phenotype), presented in the algorithm as permutation (genotype). Positions of permutations represent genes.

The number of individuals of the population is denoted by $N$ and remains constant during the search. Initial population includes solutions generated randomly or by means of constructive heuristics. The main constructive heuristic is the following: a random non-scheduled job is selected at each step. The position is assigned to this job such that gives the best objective for the current partial solution. Also the population contains permutations, that correspond non-decreasing order of due dates $d_j$, non-increasing order of $w_j/p_j$ and non-increasing order of $(w_j/p_j) \cdot exp\{- \max\{0, (d_j - p_j)\}/(n * p_{aver})\}$, where $p_{aver} = \sum_{j=1}^{n} p_j/n$.

Individuals of the last population is improved by local optimization procedure based on the *swap* neighborhood (positions of two jobs are exchanged) and the *insert* neighborhood (a job is inserted in some other position). For reducing the running time of local search heuristics, we use strategies provided in Geiger M.J. (2010) specially for the case of the problem $1|r_j = 0, d_j, w_j| \sum_j w_j T_j$. In particular, obviously unpromising moves are excluded and only a part of the neighborhood is considered by moving jobs only from positions located at a distance of no more than $20\%n$ from each other.

---

**Algorithm 1** Hybrid Evolutionary Algorithm with Steady State Replacement Scheme.

1 Construct the initial population.
2 Until a termination condition is satisfied, perform steps
    2.1 Select two parent solutions $\pi^1$ and $\pi^2$.
    2.2 Apply mutation operator to parent solutions.
    2.3 Generate an offspring $p'$ by applying a crossover operator to $\pi^1$ and $\pi^2$.
    2.4 Offspring $p'$ replaces the worst individual of the population.
3 Perform local improvements of individuals of the last population.

---

We use two mutation operators that perform a random jump within swap or insert neighborhood Geiger M.J. (2010). The operators are used for mutation with equal probability. The mutation is applied with probability $p_{\mathrm{mut}}$. In the crossover operator we solve the Optimal Recombination Problem (I)-(II) with probability $p_{\mathrm{cross}}$ (this crossover is called *OCX*, Optimized Cycle Crossover), and the well-known *OX* (Ordered Crossover) Kellegöz et al. (2008) with probability $1 - p_{\mathrm{cross}}$. Arguments for selecting such combination of the crossover operators are provided in Section 4.

One of the perspective approaches to solve the considered TWTP problem is local optimization (Variable Neighbourhood Search, Variable Neighbourhood Descent, Tabu Search, Dynasearch and others) Wang & Tang (2009); Geiger M.J. (2010); Yahyaoui et al. (2013); Fu & Chung (2016); Congram et al. (2002); Chunga et al. (2017); Bilge et al. (2007); Ding et al. (2017). The Optimal recombination may be also considered as a best-improving move in a position-based neighbourhood defined by two parent solutions.

The classic restating rule is used for balancing evolutionary search and local search: The hybrid EA performs initially $N/2$ iterations without restricted conditions, after that, EA is restarted as soon as the current iteration number becomes twice the iteration number when the best solution was obtained.

## 4 COMPUTATIONAL EXPERIMENT

In this section, the proposed evolutionary algorithm HEA-OCX is compared to the heuristics which previously demonstrated their competitiveness to the most advanced special-purpose metaheuristics for the TWTP known at that time:

Genetic algorithm (GA) from Avci et al. (2003) is the population-based algorithm with local improvements, where initial population is generated randomly; at iterations some biased randomly

| algorithm | time, sec. | $n_{hit}$ | $n_{opt}$ | $\Delta_{aver}$ |
|-----------|-----------|-----------|-----------|-----------------|
| GA | 29.11 | – | 100% | 0 |
| PVNS | 6.19 | 100% | 100% | 0 |
| m-VNS | 0.908 | 100% | 100% | – |
| HEA-OCX | 0.05 | 100% | 100% | 0 |
| HEA-DS | 0.003 | 100% | 100% | 0 |

Table 1: Results for series with $n = 40$.

| algorithm | time, sec. | $n_{hit}$ | $n_{opt}$ | $\Delta_{aver}$ |
|-----------|-----------|-----------|-----------|-----------------|
| GA | 41.02 | – | 99.2% | 0.000 |
| PVNS | 12.15 | 100% | 100% | 0 |
| m-VNS | 1.62 | 100% | 100% | – |
| HEA-OCX | 0.14 | 100% | 100% | 0 |
| HEA-DS | 0.006 | 100% | 100% | 0 |

Table 2: Results for series with $n = 50$.

chosen subset of jobs is fixed in the selected solution and local improvements are applied on the rest jobs with adopting priorities (it is tested on a computer with Pentium II 400MHz, 96Mb).

Population-based VNS algorithm (PVNS) Wang & Tang (2009) uses the scheme of the population-based variable neighborhood search method with shaking procedure based on insertion and swap neighbourhoods and such local search techniques as Variable depth search, Path-relinking and Tabu search; initial population is constructed by NEH-heuristic with random sequences of jobs and by deterministic rules (it is tested on a computer with Pentium IV 3.0GHz, 512Mb).

Multiple VNS algorithm (m-VNS) Chung et al. (2016) is also correspond to the basic principle of VNS, where the initial solution is generated using the dispatching rule or the roulette wheel rule; the classic insertion and swap neighbourhoods and their compound versions with series of independent moves are used in local search, specific disturbing and matching operations are applied for selecting neighbourhoods at iterations (it is tested on a computer with Intel Core 2.3GHz, 2Gb).

The hybrid evolutionary algorithm (HEA-DS) from Ding et al. (2017) represents the steady state genetic algorithm, where the initial population is generated randomly, position-based crossover and perturbation by independent swaps compose reproduction operators, Dynasearch based on swap moves is applied at the initialization stage and EA iterations for local improving solutions (it is tested on a computer with Xeon E5440 2.83GHz, 16Gb). Recall that Dynasearch is a neighborhood search algorithm, where the exponential-sized neighbourhoods are searched using dynamic programming.

We use the TWTP instances from OR-Library (http://people.brunel.ac.uk/ mastjjb/jeb/orlib/wtinfo.html). The number of jobs in the series of instances are 40, 50 and 100. Each series contains 125 tests. The Algorithm HEA-OCX was run on each instance for 50 times. The algorithm is set to stop when it obtains an optimal solution or reaches a maximum number of generations – 6000 (such condition is selected in order to compare our results to the ones form Avci et al. (2003); Wang & Tang (2009); Chung et al. (2016); Ding et al. (2017)). We set $p_{\mathrm{mut}} = 0.15$, $p_{\mathrm{cross}} = 0.8$, and use the tournament selection with the tournament size $s = 5$. The experiment was carried out on a computer with Intel Core i3-10100F CPU 3.60 GHz, 16 Gb.

| algorithm | time, sec. | $n_{hit}$ | $n_{opt}$ | $\Delta_{aver}$ |
|-----------|-----------|-----------|-----------|-----------------|
| GA | 118.97 | – | 66.4% | 0.02 |
| PVNS | 183.47 | 100% | 100% | 0 |
| m-VNS | 5.845 | 99.84% | 100% | – |
| HEA-OCX | 1.5 | 100% | 100% | 0 |
| HEA-DS | 0.032 | 100% | 100% | 0 |

Table 3: Results for series with $n = 100$.

The results of the experiment is provided in tables 1, 2, 3. Here $n_{hit}$ is the average percentage number of optimal or best known solution values found for an instance out of the given trial runs, $n_{opt}$ is the percentage of instances, where the optimum is found, $\Delta_{aver}$ is the average relative deviation from the optimum. We see that our algorithm HEA-OCX demonstrates competitive results, finding the record in all cases within comparable time. Note that we know from the previous research that the dynasearch method based on the swap-neighbourhood demonstrates leading results on the considered here series of instances in the context quality of solutions and time resources. Our algorithm HEA-OCX shows slightly worth results than HEA-DS in the term of time for reaching of the record. But we think that our time is also good from the practical point of view. Our current goal is to estimate the rationality of solving the ORP in EAs. Further research can be undertaken to using parallelized and/or approximate methods in optimized crossovers.

## 5 CONCLUSION

We proposed the hybrid evolutionary algorithm with optimized operators for the total weighted tardiness scheduling problem on one machine. The NP-hardness of the Optimal Recombination Problem was proved. An experimental evaluation on OR-Library instances indicates that the proposed approach gives competitive results.

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
