# OpenReview forum: "Hybrid Evolutionary Algorithm with Optimized Operators for Scheduling on Permutations"
_mathai.club/MathAI/2026/Conference — 2026 Oral_

### Official Review · Reviewer_TLVi · 2026-03-12
**Review: Hybrid Evolutionary Algorithm with Optimized Operators for Scheduling on Permutations**

**Rating:** 7
**Confidence:** 2

**Review:**

The paper aligns with the conference theme.

The authors propose a new hybrid evolutionary algorithm with optimized operators for a permutation-based scheduling problem (minimizing the total weighted tardiness on a single machine). In Theorem 1, the authors show that the optimal recombination problem subproblem in the proposed algorithm is NP-hard. The authors provide a verbal description of the algorithm and conduct numerical experiments on test instances from the OR-Library.
In formulating the problem, the authors use both classical literature (pre-2015) and fairly recent studies, including works by Tinos et al. (2020) and Eremeev & Kovalenko (2020).

# Comments and Corrections:

*   Reference list (general recommendation: add a DOI or another identifier, as it is currently missing from some articles).
* Lines 417-419: The reference "T.P. Chunga, Q. Fua, C.J. Liaob, and Y.T. Liub. On hybrid evolutionary algorithms for scheduling problem with tardiness criterion. Engineering optimization, 49(7):1133–1147, 2017" is cited. This article does not exist. The authors likely confused the titles during editing. The correct reference should be: "Multiple-variable neighbourhood search for the single-machine total weighted tardiness problem". Full details: Chung, T. P., Fu, Q., Liao, C. J., & Liu, Y. T. (2017). Multiple-variable neighbourhood search for the single-machine total weighted tardiness problem. *Engineering Optimization*, *49*(7), 1133–1147. DOI: 10.1080/0305215X.2016.1235707.

* Lines 441-443: The reference "On hybrid evolutionary algorithms for scheduling problem with tardiness criterion" is incorrect. The correct reference is: Q. Fu and T. -P. Chung, "A new approach for solving single machine total weighted tardiness (SMTWT) problem," 2016 IEEE International Conference on Industrial Engineering and Engineering Management (IEEM)*, Bali, Indonesia, 2016, pp. 438-441, DOI: 10.1109/IEEM.2016.7797913.

# Conclusion:
The work is quite interesting. I am not an expert in this field, so my comments are purely technical.

---

### Official Review · Reviewer_4kef · 2026-03-12
**Hybrid Evolutionary Algorithm with Optimized Operators for Scheduling on Permutations**

**Rating:** 7
**Confidence:** 3

**Review:**

This paper is theoretically grounded and shows proves authors' point accuracy. Key contributions of this article are proof of the NP-hardness of the Optimal Recombination Problem (ORP) for described setting and solution of it via bipartite graph matching.

Pros:
-Profound maths description
-Fair results presentation

Cons:
-Presented method seems to not outperform other SOTA methods by only one presented metric. I think there could be another metrics which could show method superiority in other setups.
-Literature: Lacks references to recent (2023-2025) ML-based combinatorial optimization methods.
-Scalability: Limited to n=100; exponential complexity of ORP may hinder larger instances.

Conclusion:
There is place for futher improvement, but presented results are still solid, accept.

---

### Official Review · Reviewer_3Po8 · 2026-03-12
**The review of "Hybrid Evolutionary Algorithm with Optimized Operators for Scheduling on Permutations"**

**Rating:** 6
**Confidence:** 4

**Review:**

This paper studies the single-machine total weighted tardiness scheduling problem and proposes a hybrid evolutionary algorithm with optimized recombination operators. The algorithm incorporates greedy initialization, adaptive mutation, and local search heuristics, while the crossover operator is based on solving the optimal recombination problem. The authors also analyze the computational complexity of this recombination problem and provide a constructive exact method based on perfect matchings in a bipartite graph. Experimental results on standard OR-Library benchmarks demonstrate that the proposed algorithm achieves competitive results compared with several established heuristic approaches.

Strengths:
- The paper presents a clear mathematical formulation of the scheduling and recombination problems.
- The NP-hardness analysis of the optimal recombination problem provides useful theoretical insight.
- The proposed optimized crossover operator is an interesting integration of exact combinatorial reasoning into evolutionary algorithms.
- The hybrid evolutionary algorithm combines several effective optimization components including local search and adaptive mutation.
- Experimental results on benchmark instances indicate competitive performance of the proposed approach.

Suggestions for improvement:

The paper could be strengthened by:
- providing additional analysis of the algorithmic behavior of the evolutionary framework;
- expanding experimental evaluation with additional benchmark instances and statistical comparisons;
- clarifying the practical implications of optimized recombination for other permutation-based optimization problems.

Final Recommendation:

POSTED / Poster-style acceptance with revision

Overall, the paper presents a technically sound contribution to permutation-based optimization and evolutionary scheduling algorithms. The combination of theoretical analysis and empirical evaluation makes the work suitable for discussion within the MathAI community.

---

### Decision · Program_Chairs · 2026-03-14

**Decision:**

Accept (Oral)

**Comment:**

Dear Author(s),

On behalf of the Program Committee of the International Conference on Mathematics of Artificial Intelligence (MathAI 2026), we are pleased to inform you that your paper has been accepted for an oral presentation at MathAI 2026.

Your paper was evaluated through a rigorous two-stage review process involving both automated screening and expert review by members of the Program Committee. The reviewers recognized the quality and contribution of your work.

Presentation details:

- Format: Oral presentation (15–20 minutes + 5 minutes Q&A)
- Mode: You may present either in person (offline) at the conference venue in Sirius, Russia, or remotely via Zoom. Please indicate your preferred mode when confirming your participation.
- Conference dates: Marh 30 - April 3, 2026
- Website: https://mathai.club

Next steps:

1. Please confirm your participation and presentation mode by replying to this email mathai.club@yandex.ru no later than March 15, 2026 18:00 Moscow time.
2. If you plan to attend in person, the organizing committee will provide accommodation details separately.
3. Please prepare your final camera-ready manuscript according to the formatting guidelines available at https://mathai.club and upload it to OpenReview by March 15, 2026 18:00 Moscow time.

Should you have any questions regarding the program, logistics, or your presentation slot, please do not hesitate to contact us.

We look forward to your contribution to MathAI 2026.

With kind regards,

MathAI 2026 Program Committee
International Conference on Mathematics of Artificial Intelligence
https://mathai.club
OpenReview: https://openreview.net/group?id=mathai.club/MathAI/2026/Conference
Telegram: https://t.me/MathAI_club
Email: mathai.club@yandex.ru